# Microstructure and Mechanical Properties of Steel and Ni-Based Superalloy Joints for Rotors of High-Speed Electric Motors

**DOI:** 10.3390/ma15196906

**Published:** 2022-10-05

**Authors:** Eero Scherman, Eerik Sikanen, Hemantha Kumar Yeddu, Mohsen Amraei, Jussi Sopanen

**Affiliations:** 1Mechanical Engineering, LUT School of Energy Systems, LUT University, P.O. Box 20, 53851 Lappeenranta, Finland; 2Department of Technology, LAB University of Applied Sciences, Yliopistonkatu 36, 53850 Lappeenranta, Finland; 3Department of Mechanical and Materials Engineering, University of Turku, 20520 Turku, Finland

**Keywords:** iron alloys, nickel alloys, shear strength, microstructure, grains and interfaces, finite element analysis

## Abstract

High-speed electric motors, e.g., axially laminated anisotropic synchronous reluctance motors (ALA-SynRM), use a solid rotor manufactured by joining alternating layers of magnetic and non-magnetic metallic sheets. The strength of the dissimilar metallic joints is critical for the rotor’s ability to withstand the operating conditions of the high-speed electrical machine. In this work, various dissimilar metallic joint configurations that can be used in high-speed ALA-SynRM rotors are studied by analyzing the shear strength, microstructure, hardness, and composition of the joints. Metallic joints of structural steels and Inconel^®^ alloys fabricated by vacuum brazing and hot isostatic pressing (HIP) are studied. Finite element analysis (FEA) was performed to calculate the maximum shear stress of the joints that were subjected to various high speed operating conditions. The shear strength of the test specimens was measured and compared with FEA results. The microstructure and chemical composition of the joints were studied by using optical microscopy, scanning electron microscopy (SEM) and energy dispersive spectroscopy (EDS) on SEM. The results show that the hot isostatic pressed S1100MC-IN718 joint achieved the highest ultimate shear strength (233.3 MPa) followed by vacuum brazed S355MC-IN600 joint (230.1 MPa) and HIP S355-IN718 (203.5 MPa), thereby showing that vacuum brazing and HIP can be viable manufacturing methods to fabricate a high-speed ALA-SynRM rotor.

## 1. Introduction

High-speed (HS) electrical machine (EM) technologies offer a viable solution for the needs of future electromechanical energy conversion systems. A key benefit of high-speed technology is its higher power density, which decreases the size of high-speed systems and reduces the need of raw materials. Moreover, high-speed machines exhibit greater efficiency than electrical machines with lower rotational speeds [1]. Typical applications for HS-EM are, for example, gas turbines, flywheel energy storage systems, high-speed spindle applications, turbomolecular pumps, gas compressor applications, as well as industrial air compressors and air blowers [2]. An important aspect of electromagnetic design and manufacturing of such high-speed technology is rotor dynamics, which influences robustness and efficiency, and consequently the economics of the rotating electrical machine. High-speed rotating electrical machines are electrical machines with rotor surface speed greater than 100 m/s [3] and high rotational speed with respect to the power of the electrical machine [2]. The maximum operation speed of a HS-EM is governed by the mechanical robustness of its rotor at high speeds [4], thermal losses in the rotor and in the stator occurring at high switching frequencies [5], and gas friction losses at rotor surface speed over 300 m/s [6].

Typically, high-speed electrical machines can be classified by machine type and rotor structure into induction machines (IM), permanent magnet synchronous machines (PMSM) and switched reluctance machines (SRM) [7]. Synchronous homopolar machines [8] and synchronous reluctance motors (SynRM) [9] are also applicable to be used in high-speed applications.

Each of the high-speed electrical machine types listed above has a unique rotor structure which makes the machine particularly suitable for specific application. The strengths and weaknesses of the various high-speed electrical machine types are described in [4]. As a summary, it can be concluded that induction machines are generally suited for high-speed applications due to the mechanical robustness of the IM rotor. However, the efficiency of the IM rotor is decreased by rotor joule losses. Switched reluctance machine rotors are simple and robust constructions, but the requirement of a large airgap reduces their efficiency. Permanent magnet synchronous machines have high power density and efficiency; however, the mechanical robustness of the rotor, which is affected by the magnet encapsulation, may be a limiting factor in high-speed applications.

To overcome some of the limitations of conventional high-speed rotors noted above, an axially laminated anisotropic synchronous reluctance motor (ALA-SynRM) was presented in [7]. The ALA-SynRM rotor is a solid rotor structure made from multiple alternating magnetic and non-magnetic metal sheets that provide the necessary flux paths for the electromagnetic field. A key benefit of the ALA-SynRM rotor is its simplicity. Unlike PMSM rotors, the ALA-SynRM rotor does not require rare earth permanent magnets. Moreover, there is no need for the complicated squirrel-cage structure found in IM rotors, which can limit the maximum rotational speed of the rotor by decreasing the rigidity of the rotor [10].

Typical methods for bonding dissimilar metals sheets includes brazing and high and low temperature solid-state welding processes, such as explosion welding and hot isostatic pressing (HIP), all of which can be used to manufacture ALA-SynRM rotors [11]. The brazing process is described as a process where the parent materials to be bonded are heated to a temperature that is lower than the melting temperature of the parent materials but higher than the melting temperature of the braze alloy [12]. As the braze alloy melts in the joint, capillary forces and the wetting action cause the braze alloy to flow throughout the joint area and, on cooling, to create a permanent bond between the workpieces. In the explosion welding process, an explosive charge is used to accelerate the parts to be bonded to high speeds of up to several hundreds of meters per second. The parts collide with each other and solid bonding of the two metal pieces occurs at the point of impact [13]. The HIP process is used as a solid-state diffusion welding process to bond dissimilar materials and to sinter powder metal components with fully isotropic material parameters [14]. The joint is formed by deforming and diffusing the joint area using high temperature and pressure [15].

Recently, the electromagnetic properties of ALA-SynRM rotors manufactured by mechanically bonding the laminated rotor structure have been studied in [16,17,18] and all aforementioned researchers highlight similar or higher rotor efficiency compared to other rotor structures. These studies, however, consider only ALA-SynRM rotors manufactured by mechanically joining, e.g., with screws, the alternating magnetic and non-magnetic metallic sheets. Using external elements to construct the ALA-SynRM rotor can lead to a complicated construction process in terms of manufacturing and may compromise the structural integrity of the rotor in high-speed applications [19]. The utilization of metal-to-metal bonding methods, i.e., brazing, explosion welding and hot isostatic pressing, to manufacture ALA-SynRM rotor structures has not been widely studied. While the feasibility of a brazed ALA-SynRM rotor was studied in [20], it was primarily focused on describing the electromagnetic performance of the brazed ALA-SynRM rotor with little consideration given to joint strength and structure.

The detailed analysis of EM rotors structural performance and its applicability in high-speed operations in [21] highlighted the importance of effective bonding of the dissimilar materials. Accordingly, this article investigates dissimilar metal joints suitable for use in construction of the rotor of an ALA-SynRM. The mechanical properties, such as shear strength, and microstructural aspects of the joints made from several different materials are studied. The paper begins by giving an overview of the structure of an ALA-SynRM rotor. The subsequent section introduces the experimental methods used to determine the shear strength of the joints followed by finite element analysis of the stresses that the rotor is subjected to during its operation. Finally, the shear strength measurements and the corresponding microstructural aspects are discussed. The paper concludes by reprising key findings from the work.

## 2. Materials and Methods

During operation of an electrical machine, the rotor is subjected to mechanical loads caused by the rotational motion of the rotor and electromagnetic forces created by the electrical machine. Additionally, the rotor may be subjected to thermal loads due to varying operational temperature. The mechanical robustness of the rotor is often the limiting factor for high rotational speeds.

In the case of an ALA-SynRM rotor, the rotor is constructed from multiple magnetic and non-magnetic sheets bonded to each other. A two-pole ALA-SynRM rotor cross section is shown in Figure 1. In the ALA-SynRM rotor, the magnetic sheets act as a flux path and the non-magnetic sheets act as insulators that guide the electromagnetic flux [22]. The strength of the joint combining the sheets depends on the strength of the parent material and the joint integrity which are studied using experiments and simulations as explained in the following sections.

### 2.1. Experiments

#### 2.1.1. Materials

This study focuses on the strength and structure of the following three bimetallic joint combinations: (i) vacuum brazed joint of structural steel S355MCD and non-precipitation hardened Inconel^®^ 600, (ii) hot isostatically pressed structural steel S355MC and precipitation hardened Inconel^®^ 718, and (iii) ultra-high strength structural steel S1100MC and precipitation hardened Inconel^®^ 718. Material pairs for this research were selected based on existing literature [23], as well as the aim of studying the possibilities of using ultra-high strength structural steel in the construction of an ALA-SynRM rotor. The chemical composition of the parent materials and braze alloy for vacuum brazing of this study are presented in Appendix A Table A1.

Table 1 presents the thickness, yield strength, ultimate tensile strength, elongation, hardness, and delivery state of the materials and braze alloy studied at a temperature of 20 °C in the delivery state as indicated in the material certificates. For S1100, Inconel^®^ 600, Inconel^®^ 718, and braze alloy CW021A, yield strength is 0.2% proof stress (Rp0.2).

#### 2.1.2. Bonding Processes

Figure 2 shows the shape of the billet used to construct the ALA-SynRM rotor and the shape of the test specimens extracted from the rotor billet.

Table 2 describes the bonding methods and test specimens used in this study. Test specimens to study HIP bonding were manufactured from the excess material of the billets used to manufacture the ALA-SynRM rotors (Figure 2). To study the vacuum brazed joint, a separate test billet was manufactured after the manufacturing of the rotor billet, since the rotor billet did not provide enough test specimens to study the joint. The vacuum brazed test specimen contains only thick magnetic sheets to provide a more favorable test procedure in practice.

Control specimens D, E, F, G and H were included in the experiment. Specimen D was used to validate the shear strength test setup by measuring the shear strength of a parent material with known strength properties. Control specimens E, F, G and H were used to determine the tensile strength of the parent materials after heat treatment similar to the test specimens used in the bonding processes.

Manufacturing of the rotor and test specimen billets began by cutting the metal plates to suitable shapes with a fiber laser. After preparation, the cut plates were sent to companies specializing in the vacuum brazing and hot isostatic bonding processes. For test specimen A, the rate of heating in the brazing process was 314 °C/h before 30 min dwell time at a brazing temperature of 1100 °C. In the brazing atmosphere, the pressure was less than 0.0001 MPa during the brazing process. The cooling cycle lasted 2 h, during which the test piece cooled down to 950 °C in the first hour and to room temperature in the second hour. A copper braze alloy with thickness of 100 μm was utilized in the vacuum brazing while the strength and ductility of the braze alloy material was considered to be suitable on this application.

Authors in [23] suggested that HIP should be used when bonding Inconel^®^ 718 because vacuum brazing of Inconel^®^ 718 and S355 structural steel was experimented with copper braze alloy and results were unsatisfactory. Thus, the material pairs including Inconel^®^ 718 in this research are bonded by HIP. In the HIP process, the rate of heating was 164 °C/h and the rate of pressure increase was 14.4 MPa/h. The rate of cooling was 192 °C/h and the rate of pressure decrease was 16.3 MPa/h. The HIP process was performed in an argon-protected atmosphere. After successful bonding, all test specimens were visually inspected before machining to a shape suitable for the test setup.

The control specimens E and F were exposed to a high temperature of 1100 °C for a long time (5 h) to simulate the effect of long-term exposure to high temperature simulating HIP process. The control specimens G and H were exposed to a similar temperature as in the vacuum brazing for one hour to simulate the temperature cycle of vacuum brazing. All control specimens were encapsulated in individual argon protected atmospheres during exposure to high temperature.

#### 2.1.3. Experimental Testing

When assessing the strength of a joint, either the normal or the shear strength can be studied. Due to the structure of the ALA-SynRM rotor, the latter is the focus of this research. Shear stress along a lap joint can be evaluated by using test specimen geometry as demonstrated in [24] or directly induce shear stress on the test specimen with a dedicated test setup as in [25]. In this study, a test setup adopted from [25] was created and Figure 3 illustrates the test setup and the principle of operation.

In the study, the test specimen was carefully placed such that the joint was located between the upper and lower die. In each test, the structural steel was located under the upper die and, respectively, Inconel^®^ was always on top of the bottom die.

The shear strength of the joints was determined using a universal testing machine (Matertest Oy FMT-250, Finland) equipped with a servo-hydraulic cylinder with a maximum force capacity of 250 kN. Test specimens were subject to a load which was gradually increased up to the breaking point with the strain rate indicated in Table 3. The universal testing machine was equipped with a calibrated load cell (Interface Inc. 1020AF-125KN-B, USA), which allowed the force exerted to be recorded during the test. The servo-hydraulic cylinder was equipped with a position transducer (Curtiss-Wright Corp., Penny & Giles VRVT100, United Kingdom), which allowed the displacement of the die during the tests to be recorded. The universal testing machine was also used to determine the ultimate tensile strength of the control specimens E, F, G and H.

Metallographic samples were prepared and etched using 4% Nital for 10 s for structural steel samples and 10% oxalic acid with 30 V/2 A DC-current for Inconel samples. The microstructure of the specimens was examined with a stereo microscope (Meiji Techno IM7530, Japan). The microhardness of the specimens was measured with a micro hardness tester (Struers Durascan, HV3/10, Denmark). A scanning electron microscope (Hitachi SU3500, Japan) was used to study the composition of the joint area in greater detail.

### 2.2. Finite Element Simulations

The design parameters for the rotor under study are as follows: maximum rotational speed of 24,000 rpm and nominal power of 12 kW. The laminate layer thicknesses and configuration are shown in Figure 4. The materials initially considered for the finite element (FE) study of the laminated rotor are S355 steel for the magnetic material and Inconel^®^ 718 for the non-magnetic material, due to the following reasons. In addition to the excellent electromagnetic performance, this material pair has very similar structural properties, which minimizes the difference in thermal growth under thermal load and, in general, the overall strain difference between the laminated sheets. In addition, the availability and price of the materials are favorable. Finite element analysis (FEA) was performed also for other material pairs shown in Table 3. The material properties used in FEA are shown in Table 4. The software used for finite element analysis was Ansys Workbench 2021 R2.

Based on the CAD geometry depicted in Figure 4, a finite element (FE) model was generated to be able to solve the stresses the structure experiences under certain loading conditions. The rotational velocity was set to nominal speed 24,000 rpm, and the estimated thermal condition was included in the model by applying constant 300 °C increase from the initial condition. To maintain the size of the numerical problem feasible for solution, bonding material layers and the thermal loading cycle during the bonding process were excluded from the analysis. Instead, the laminated layers were directly connected using fixed constraints to simulate perfect bonding of laminated layers. Mesh density was adjusted to have a minimum of two element layers in the direction of thickness in each laminate, resulting in a total of 1,103,000 nodes in a mixed hexahedron dominant mesh having a total of 353,000 quadratic elements. This particular mesh density was utilized to optimize the computational time with the hardware resources available and it was deemed appropriate for the purposes of comparative analysis of the joints. In all the finite element analyses, the model geometry and the mesh density were kept unchanged, to be able to focus the analysis only on the behavior of different materials under the stresses.

## 3. Results

### 3.1. Experiments

#### 3.1.1. Shear Strength and Hardness

The data gathered during the shear strength tests was analyzed and used to assess the shear behavior of the bimetallic joints. The engineering value of the shear stress was determined by dividing the force exerted on the test piece by the cross-section area of the test piece. Due to the nature of the test setup, only the displacement of the upper die (Figure 3b) could be recorded from the data provided by the position transducer in the servo-hydraulic cylinder. To obtain shear stress versus the displacement of the die, the displacement was normalized with the specimen thickness similarly as [30].

Figure 5 presents the shear stress-normalized displacement for the test specimens, and Table 5 and Table 6 show the results of the shear strength of the joints and the ultimate tensile strength of the control specimens, respectively.

According to the shear strength tests, the shear strength of the vacuum brazed S355-Inconel^®^ 600 joint and S1100-Inconel^®^ 718 HIP joint is almost equal regardless of the different bonding methods and different parent materials. It is noteworthy that the joint configuration S355-Inconel^®^ 718 HIP produced the lowest shear strength of the study.

Test specimen D was used to determine the shear strength of the parent material S355MCD as a reference for the test setup. The results showed an ultimate shear strength of 300.3 Mpa (*n* = 10, SD = 13.3 Mpa). The material certificate for S355MCD showed an ultimate tensile strength of 490 Mpa (Table 1). Considering the von Mises yield criterion, the measured ultimate shear strength of the parent material is slightly higher than the theoretical value ($490\;MPa\times \frac{1}{\sqrt{3}}\;\approx 285\;MPa)$.

Figure 6a–c describe the macro structure and micro hardness profile (HV3/10) of the joint in the vacuum brazing and HIP test specimens. The microhardness profile of the S355MCD-IN600 test specimen does not exhibit major changes in the hardness of the materials compared to the hardness at delivery state. Only minor loss of hardness compared to the delivery state can be noted in the structural steel S355MCD. However, the hardness profile of the S1100MC-IN718 test specimen displays a significant change in the hardness compared to the delivery state of the parent materials. The hardness of Inconel^®^ 718 shows high increase in hardness compared to the parent material in the delivery state and the hardness of structural steel S1100MC is less than half of the original hardness as given in Table 1. The joint configuration of structural steel S355MC-IN718 exhibits similar behavior to the S1100MC-IN718 test specimen.

#### 3.1.2. Microstructure

Figure 7 shows the microstructures of the parent materials in the delivery state and after bonding. Additionally, average grain size ($\bar{P_{L}})$ of each parent material in the delivery state and after bonding was determined using a mean linear intercept procedure following standard SFS-EN ISO 643:2020. Due to the complex microstructure of structural steel S1100MC, the average grain size in the delivery state could not be determined by using the mean linear intercept method.

All the test specimens show considerable change in the microstructures before and after the joining processes. For structural steel S355MC, the fine-grained ferrite–pearlite microstructure has changed to a coarse-grained ferrite–pearlite microstructure following the vacuum brazing and hot isostatic bonding processes. The microstructure of the as-delivered thermomechanically formed S1100MC steel is a mixture of bainite and martensite with retained austenite islands scattered along grain boundaries [31], but after the HIP bonding, the microstructure is coarse-grained ferrite–pearlite.

The microstructure of Inconel^®^ 718 as-delivered state is solid solution austenite (γ) with minor phases γ′, γ″, and δ [32]. After the HIP bonding, the microstructure has a small grain size in the joint area and relatively large grain size farther away from the joint area. Twinning is also apparent in the microstructure, which is identified by the straight lines across the grains. Inconel^®^ 600 has an austenitic microstructure with titanium nitride and titanium carbide precipitates in the soft annealed state [28]. After the vacuum brazing, the microstructure of the Inconel^®^ 600 has a very large grain size, similar to the Inconel^®^ 718 test specimen.

Scanning electron microscopy (SEM) was used to gain a better understanding of the chemical composition of the joint; see Figure 8, Figure 9 and Figure 10. The chemical composition was determined by energy dispersive spectroscopy (EDS) using secondary electron (SE) detector. The amounts of different alloying elements, in weight percentage, in each joint is presented with a color legend.

### 3.2. Finite Element Simulations

The results of the FE analysis are shown in Figure 11 using an S355MC-Inconel^®^ 718 material pair. Maximum shear stress was selected as the parameter for study as it indicates the minimum required shear strength for the laminated joint. In Figure 11a, the simulation results show the maximum shear stress in the structure to be no greater than 37 MPa. This result defines the minimum shear strength required for the joining of two different materials used, and it will be compared against the experimental results. The maximum shear stress was selected for this comparison as it indicates the magnitude of the overall shear stress more clearly than shear stress in the *XZ*-plane under the current loading conditions. A clear difference in the maximum shear stress distribution of the two materials is seen in Figure 11a, as the non-magnetic Inconel^®^ 718 is under almost two times greater shear stress than the magnetic S355 steel. This result can be explained by the greater density of Inconel^®^ 718, and thus, a greater centrifugal load is experienced by this material. In addition, it also possesses a lower modulus of elasticity, resulting in greater strain in Inconel^®^ 718. Maximum principal stress is shown in Figure 11b. In this case, the magnetic S355 is under greatest stress having an approximate maximum principal stress of 77 MPa. This result indicates the minimum yield strength for the materials used. Additional stress results with different material pairs are listed in Table 7.

## 4. Discussion

None of the bonded test specimens A, B, and C achieved the same level of ultimate shear strength as the parent materials (Table 5). For S1100MC, the theoretical shear strength considering the von Mises yield criterion is about 790 MPa in its as-delivery state. With the same considerations, the shear strengths of Inconel^®^ 600 and Inconel^®^ 718 are about 400 MPa and 510 MPa, respectively. Therefore, the shear strength of the joints was lower than that of the parent materials, i.e., the measured shear strength of S355MCD (Table 5) and the theoretical shear strength of other parent materials in their delivery state. Moreover, it is noteworthy that the ultimate shear strength of the joint in S1100-IN718 (test specimen B) and S355-IN718 (test specimen C) is only 30 MPa higher in test specimen B although the strength of the parent material S1100 is much higher in the delivery state as compared to the S355 used in test specimen C.

In a study focused on describing the optimal vacuum brazing temperature of Inconel^®^ 600 alloy when using AgCuTi braze alloy [25], it was stated that the shear strength reached the highest value of 223.32 MPa at a brazing temperature of 865 °C if Inconel^®^ 600 is bonded by vacuum brazing. The study also drew attention to the profound effect of the process parameters, mainly brazing temperature, on the strength of the joint. It is noteworthy that the shear strength of 230.1 MPa of the vacuum brazed S355-IN600 joint shows that the shear strength of IN600 is not affected significantly by joining with S355 steel, although the braze alloy compositions and brazing process variables in the present study and in [25] are different. In [23], the ultimate tensile strength for a vacuum brazed S355J0-Inconel^®^ 600 joint was found to be 283 MPa. Vacuum brazing Inconel^®^ 718 was studied in [33] by examining γ-TiAl alloy/Inconel^®^ 718 vacuum brazed joints. In the work, it is claimed that vacuum brazing IN718 is possible with a suitable braze alloy. The braze alloy used contained silver, copper, indium, and titanium, and the average shear strength achieved in the study was 228 ± 83 MPa at a brazing temperature of 730 °C.

The shear stress-normalized displacement curves (Figure 5) showed that there is some variation in the ductility of the joint since test specimen C shows different behavior compared to test specimens A and B. The behavior of test specimen C changes from linear elastic behavior to plastic region at lower shear stress. Overall, the ductility of test specimen C is higher than the ductility of test specimens A and B, as rupture occurred at a higher plastic strain. However, the ultimate shear strength of test specimen C is lower compared to other test specimens. These findings need thus to be interpreted with caution since this behavior could not be verified by actual measurement of the yield point with the test setup used in this study. The failure of the specimens observed in Figure 5 can be further confirmed by the study of the fracture surfaces (Appendix B). By observing the fractured specimens, it can be stated that test specimen A has failed by brittle fracture which is indicated by the very flat fracture surface. Test specimen B has failed by mixed fracture mode, which is noticeable by mildly slanted edges and partly flat fracture surface in the center of the specimen. Observing fracture surface of test specimen C, it can be stated that the failure mode is ductile failure due to the highly slanted edges of the fractured specimen.

The FE analysis described in Section 3 indicates a maximum permissible shear stress not to be greater than 37 MPa using the initial S355MC-Inconel^®^ 718 material pair in the studied ALA-SynRM rotor structure at the nominal rotational speed and at operational temperature. Other material pair with S1100 and Inconel^®^ 600 (material pairs A and B) yields greater shear stresses, although no greater than 60 MPa. It is noticeable in Table 7 that the location of maximum stresses changes between the magnetic center laminate layer and non-magnetic laminate layer with different material pairs, as illustrated for S355MC-Inconel^®^ 718 pair in Figure 11. Magnetic steels S355 and S1100 have very similar material properties, as indicated in Table 4. The similar stress distributions in the laminate layers with material pairs B and C can be explained by the presence of Inconel^®^ 718, as indicated in Table 7. In all material pairs, the material with a lower coefficient of thermal expansion is under the greatest tensile stress. Since the relative change in the density is smaller than the relative change in the coefficient of thermal expansion, it can be concluded that the main contribution to the simulated stresses is due to thermal strain. The shear strength values from the experimental tests support vacuum brazing and hot isostatic pressing methods as suitable bonding processes for ALA-SynRM rotors since both methods can achieve higher shear stress strength than the value predicted by FE analysis.

The microstructure and composition of the S355MC-IN600 vacuum brazed joint shows little or no diffusion of the main elements of the respective parent materials to their counterparts across the joint interface (Figure 8a,c). Additionally, a distinctive copper layer is present because of the copper braze alloy (Figure 8b). Manganese (Figure 8e) and silicon (Figure 8f) show good diffusion to the copper braze alloy. In addition, nickel shows some diffusion to the copper braze alloy (Figure 8c) whereas chromium (Figure 8d) and iron (Figure 8a) have slightly less diffusion to the copper braze alloy. A thin line of a nickel (Figure 8c) and chromium (Figure 8d) enriched area has also formed near the joint face of the S355MC. Additionally, on the joint face of IN600, precipitates of Cr and Cu have formed as indicated by the strong presence of chromium and copper in the same location in which dark spots are visible in the micrograph (arrow in Figure 8a). These Cr-Cu precipitates form alongside γ′ (Ni_3_Al, Ni_3_Ti) precipitates, whose presence is confirmed by the precipitate-like morphologies seen in the Ni-rich region (arrow in Figure 8d).

In the S1100MC-IN718 joint, significant levels of aluminum, titanium, and niobium (Figure 9d–f), which are the main elements of minor phases, viz. γ′ (Ni_3_Al, Ni_3_Ti) and γ″ (Ni_3_Nb) precipitates [32], are seen along the joint face of IN718, indicating the formation of these precipitates. γ′ and γ″ precipitates form in the joint area almost along a straight line as seen in Figure 9d–f. The dark precipitates in S1100MC near the joint could be MnS and SiC, based on the high intensity of Mn and Si visible in the respective composition plots (Figure 9h,i).

In the S355MC-IN718 joint, the main elements of each parent material show very little diffusion in the joint area (Figure 10a–c) and some aluminum, titanium, and niobium enriched areas are present in the form of γ′ and γ″ precipitates. Γ’ precipitates form along a straight line near the joint surface more prominently than the γ″ precipitates (Figure 10d–f).

The results in this study show that the long high temperature exposure during bonding processes affects the microstructure of the materials in a significant manner. The long exposure to temperatures above the recrystallization temperatures has caused changes in the microstructure and mechanical properties. The structural steels (S355MC, S1100MC) had reduced ultimate strength, whereas the ultimate strength of the Inconel^®^ grades was higher than the values specified by the manufacturers in the delivery state (Table 6).

All materials have exhibited grain growth due to the high temperature exposure. For example, for Inconel^®^ 600, grain growth begins at a temperature of 980 °C by coalescing carbides in the microstructure. Exposure for 1–2 h at temperatures of 1090–1150 °C will completely dissolve carbides and result in an increased grain size [28].

As the HIP process is carried out at 1150 °C for 5 h (Table 2), high strength phases such as martensite and bainite revert to austenite, which then transforms to ferrite and pearlite during the slow cooling of the joint. Moreover, at such high temperatures, significant grain growth occurs. The sharp drop in hardness of S1100MC after the joining process (Figure 6b) is attributed to the large austenite grain sizes and change in phase composition, i.e., from high strength and hard phases such as martensite and bainite to a relatively soft coarse-grained ferrite–pearlite mixture. Despite the loss of the lath martensitic structure due to the long-time high temperature exposure during HIP, the shear strength of the S1100MC-IN718 joint is slightly higher than the shear strength of the S355MC-IN718 joint and almost the same as the strength of the S355MC-IN600 joint.

Although the HIP process temperatures for S1100-IN718 and S355-IN718 joints are the same (1150 °C), the dwell time of the S1100-IN718 joint is longer compared to the other joint (Table 2). This longer process time used for the S1100-IN718 joint enables the diffusion of alloying elements, such as Al, Nb, Ti, much closer towards the joint region. This gives rise to the formation and concentration of γ′ and γ″ precipitates very near to the joint region rather than in regions that are slightly farther away from the joint (Figure 9). Consequently, this leads to depletion of alloying elements (Ti, Al, Nb) and a lack of evenly distributed precipitates in the regions that are slightly farther away from the joint, thereby giving rise to the observed sudden variations in the hardness of IN718 near the joint (Figure 6b). In the S355-IN718 joint, due to a relatively lower dwell time, the alloying elements, such as Al, Ti, Nb are more evenly distributed in IN718, consequently leading to even distribution of the precipitates (Figure 10d–f). Therefore, no sudden drop in hardness near the joint is observed (Figure 6c).

## 5. Conclusions

In this work, dissimilar metal joints which can be used to bond alternating magnetic and non-magnetic metal sheets to manufacture a high-speed ALA-SynRM rotor were studied. The material pairs and bonding types selected for study were vacuum-brazed S355MCD-Inconel^®^ 600 and hot isostatically pressed S1100MC-Inconel^®^ 718, and S355MC-Inconel^®^ 718. Shear stress tests were performed to determine the shear strength of the joints. Additionally, finite element analysis (FEA) of the ALA-SynRM rotor was performed to predict the required strength of the joints at the nominal rotation speed of the proposed rotor geometry. FEA showed that the material parameters (density, elastic modulus, and thermal expansion ratio) affect the stresses that the rotor experiences during operation. Material pair S355-IN718 generated the lowest stresses in this study. FEA showed that the joints should have a shear strength of at least 60 MPa, 60 MPa, and 37 MPa for S355-IN600, S1100-IN718, and S355-IN718, respectively. FEA also showed that the parent materials should have a minimum yield stress of at least 137 MPa, 117 MPa, and 77 MPa in S355-IN600, S1100-IN718, and S355-IN718, respectively.

In the experimental tests, the hot isostatic pressed S1100MC-IN718 joint achieved the highest ultimate shear strength (233.3 MPa) followed by the vacuum-brazed S355MCD-IN600 joint (230.1 MPa) and the HIP S355-IN718 (203.5 MPa) joint. It is noteworthy that the strength of ultra-high strength structural steels, like S1100MC, cannot be utilized when using bonding processes that require long exposure to high temperature as the heat input affects the microstructure of the parent material. However, the loss in strength and hardness of S1100MC after bonding does not seem to affect the overall ultimate shear strength of the S1100MC-IN718 joint compared to that of the S355MCD-IN600 joint. In conclusion, all test specimens of the study meet the requirements set by the operation conditions and geometry of the studied ALA-SynRM rotor. Therefore, it can be concluded that vacuum brazing and hot isostatic pressing can be considered viable bonding methods to fabricate high-speed ALA-SynRM rotors.

## Figures and Tables

**Figure 1 materials-15-06906-f001:**
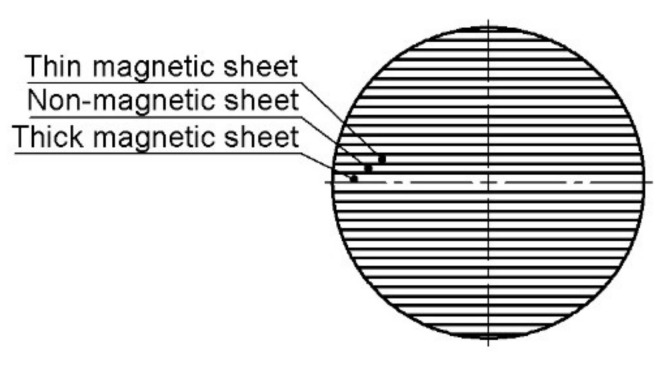
Cross section of ALA-SynRM rotor (according to [7]).

**Figure 2 materials-15-06906-f002:**
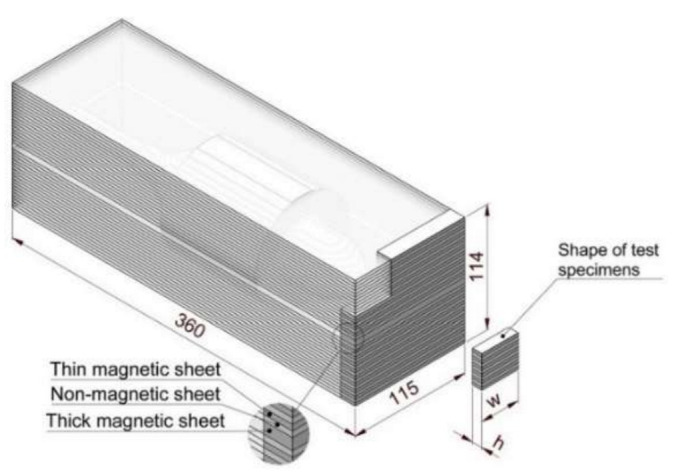
Structure of the billet for ALA-SynRM rotors and test specimens (dimensions in mm, not to scale).

**Figure 3 materials-15-06906-f003:**
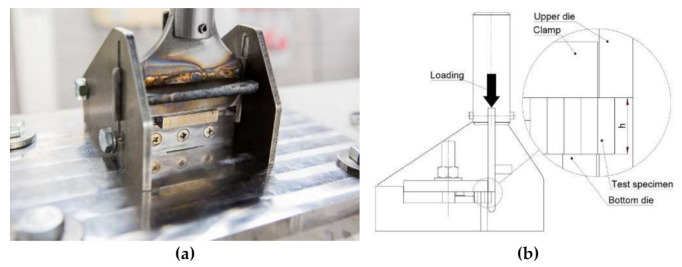
Test setup for the shear strength (**a**) experimental setup; (**b**) schematic drawing.

**Figure 4 materials-15-06906-f004:**
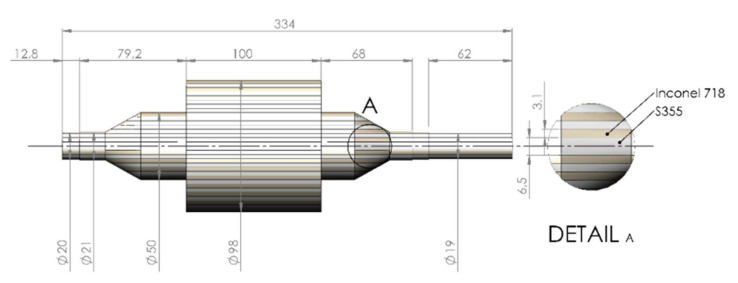
Main dimensions and laminate layer configuration of the laminated rotor. (Dimensions in millimeters, not to scale.)

**Figure 5 materials-15-06906-f005:**
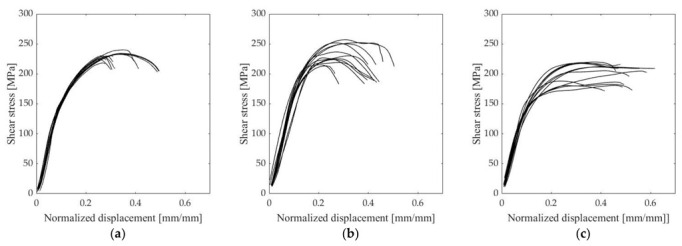
Shear stress-normalized displacement of test specimens: (**a**) S355-IN600 vacuum brazed; (**b**) S1100-IN718 HIP; and (**c**) S355-IN718 HIP.

**Figure 6 materials-15-06906-f006:**
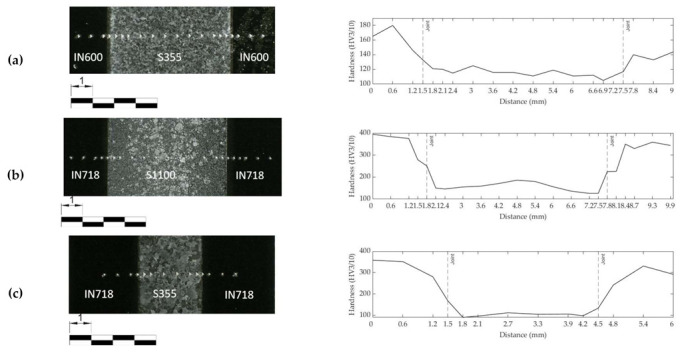
Macroscopic structure of the joints and microhardness (HV3/10): (**a**) S355-IN600 vacuum brazed; (**b**) S1100-IN718 HIP; and (**c**) S355-IN718 HIP.

**Figure 7 materials-15-06906-f007:**
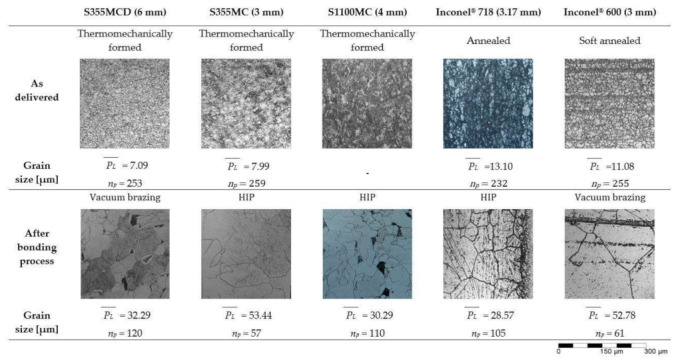
Microstructures of the test specimens (*n_p_* = number of intersections).

**Figure 8 materials-15-06906-f008:**
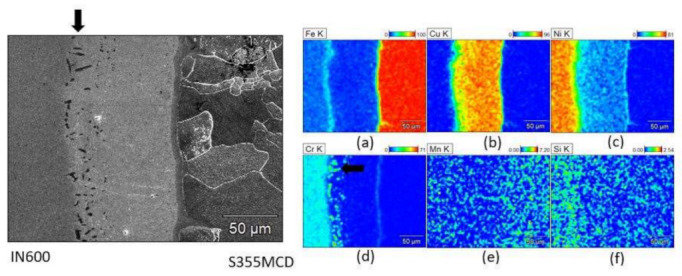
Micrographs of the structure and composition in weight % of the S355MCD-IN600 joint area: (**a**) iron (**b**) copper (**c**) nickel (**d**) chromium (**e**) manganese (**f**) silicon.

**Figure 9 materials-15-06906-f009:**
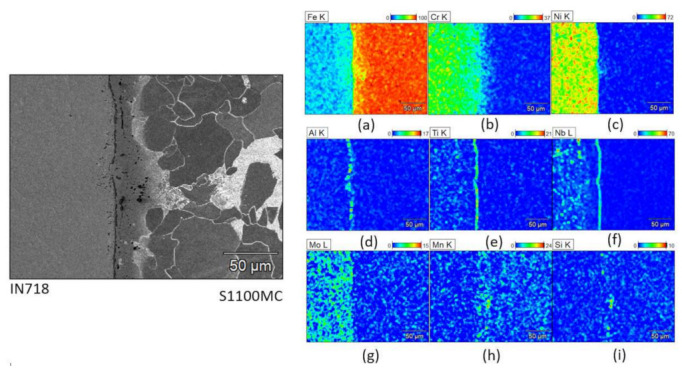
Micrographs of the structure and composition in weight % of the S1100MC-IN718 joint area: (**a**) iron (**b**) chromium (**c**) nickel (**d**) aluminium (**e**) titanium (**f**) niobium (**g**) molybdenum (**h**) manganese (**i**) silicon.

**Figure 10 materials-15-06906-f010:**
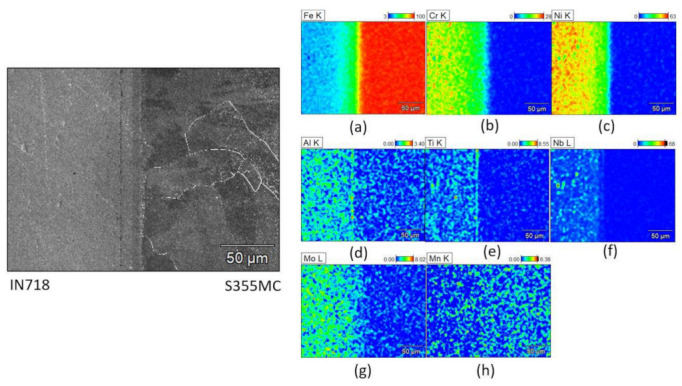
Micrographs of the structure and composition in weight % of the S355MC-IN718 joint area: (**a**) iron (**b**) chromium (**c**) nickel (**d**) aluminium (**e**) titanium (**f**) niobium (**g**) molybdenum (**h**) manganese.

**Figure 11 materials-15-06906-f011:**
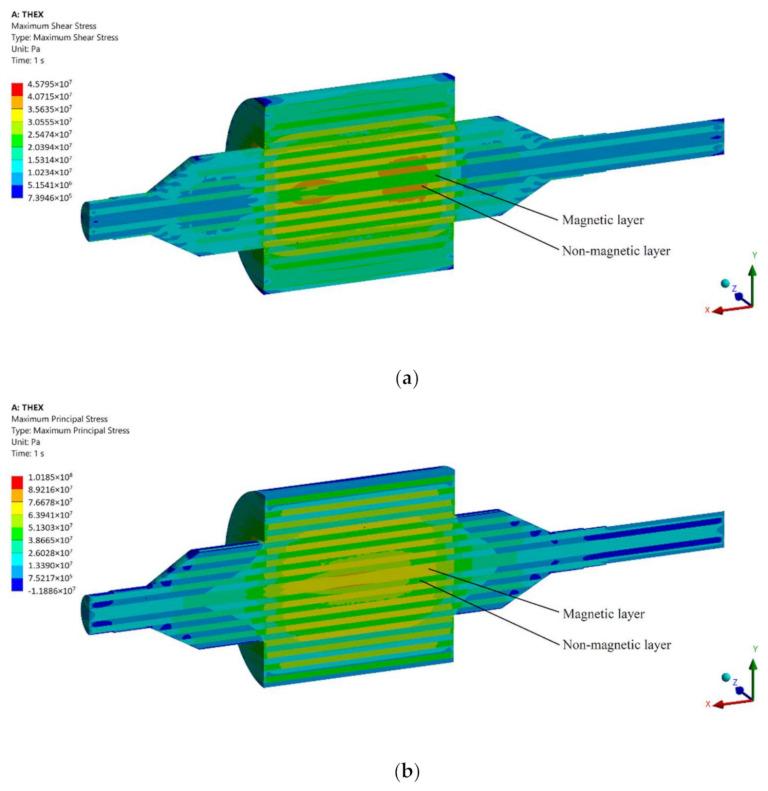
Stress distributions of laminated rotor structure under centrifugal and thermal loads. (**a**) Maximum shear stress distribution. (**b**) Maximum principal stress distribution.

**Table 1 materials-15-06906-t001:** Mechanical properties of the parent materials based on material certificates.

Material	Thickness (mm)	Yield Strength (MPa)	Tensile Strength (MPa)	Elongation (%)	Hardness	Delivery State
S355MC	3.00	402	487	34 (A45)	155 (HV3/10) ^1^	Thermomechanically formed
S355MCD	6.00	421	490	34 (A50)	155 (HV3/10) ^1^	Thermomechanically formed
S1100MC	4.00	1166	1363	9.0 (A5)	453 (HV3/10) ^1^	Thermomechanically formed
S1100MC	6.00	1144	1374	9.0 (A5)	453 (HV3/10) ^1^	Thermomechanically formed
Inconel^®^ 600	3.00	327	691	48.1 (A50)	86.2 (HRBW)	Soft annealed
Inconel^®^ 718	3.17	419	878	53	92 (HRBW)	Annealed
Inconel^®^ 718	3.71	450.2	903	50.4	95.3 (HRB)	Annealed
CW021A	0.100	98	247	36 (A50)	-	Soft annealed

^1^ measured value (Struers Durascan, HV3/10, Denmark).

**Table 2 materials-15-06906-t002:** Test specimens and their bonding methods and process parameters.

Test specimen	Material	Thickness (mm)	Bonding Method	Temperature (°C) and Dwell Time (h)	Pressure at Maximum Temperature (MPa)
A	S355MCD	6.00	Vacuum brazing	1100/0.5	0.0001
	Inconel^®^ 600	3.00			
B	S1100MC	4.00, 6.00	Hot isostatic pressing	1150/5	101
	Inconel^®^ 718	3.17			
C	S355MC	3.00, 6.00	Hot isostatic pressing	1150/4	101
	Inconel^®^ 718	3.17			
D	S355MC	6.00	-	-	-
E	S1100MC	4.00	-	1100/5	-
F	Inconel^®^ 718	3.71	-	1100/5	-
G	Inconel^®^ 600	3.00	-	1100/1	-
H	S355MCD	6.00	-	1100/1	-

**Table 3 materials-15-06906-t003:** Test and test specimen details.

Test Specimen	Test Type	Material	Bonding Method	Width (mm)	Height (mm)	Gauge Length (mm)	Strain Rate (s^−1^)
A	Shear	S355MCD	Vacuum brazing	50	10	-	0.001
		Inconel^®^ 600					
B	Shear	S1100MC	Hot isostatic pressing	36	5	-	0.002
		Inconel^®^ 718					
C	Shear	S355MC	Hot isostatic pressing	50	6	-	0.002
		Inconel^®^ 718					
D	Shear	S355MCD	-	50	6	-	0.002
E	Tensile	S1100MC	-	30	4	50	0.0002
F	Tensile	Inconel^®^ 718	-	30	3.71	50	0.0002
G	Tensile	Inconel^®^ 600	-	30	3	50	0.0002
H	Tensile	S355MCD	-	30	6	50	0.0002

**Table 4 materials-15-06906-t004:** Material properties used in finite element analysis of the laminated rotor [26,27,28,29].

Material	Density (kg/m^3^)	Elastic Modulus (GPa)	Poisson’s Ratio (-)	Thermal Expansion (1/°C)
S355	7800	210	0.30	12.0× 10^−6^
Inconel^®^ 718	8190	200	0.29	12.5× 10^−6^
S1100	7850	210	0.30	11.0× 10^−6^
Inconel^®^ 600	8470	214	0.32	10.4× 10^−6^

**Table 5 materials-15-06906-t005:** Shear strength of the test specimens (*n* = number of samples).

Test Specimen	Material	Bonding Method	*n*	Mean Ultimate Shear Strength (Mpa)	Standard Deviation of the Ultimate Shear Strength (Mpa)
A	S355MCD	Vacuum brazing	12	230.1	5.5
	Inconel^®^ 600				
B	S1100MC	Hot isostatic pressing	12	233.3	15.7
	Inconel^®^ 718				
C	S355MC	Hot isostatic pressing	12	203.5	15.5
	Inconel^®^ 718				
D	S355MCD	-	10	300.3	13.3

**Table 6 materials-15-06906-t006:** Tensile strength of the control specimens (*n* = number of samples).

Test Specimen	Material	*n*	Mean Ultimate Tensile Strength (Mpa)	Standard Deviation of the Ultimate Tensile Strength (Mpa)
E	S1100MC	5	755.3	20.2
F	Inconel^®^ 718	5	1317.5	17.2
G	Inconel^®^ 600	5	740.9	30.5
H	S355MCD	5	352.7	27.1

**Table 7 materials-15-06906-t007:** Stress results of rotor made of different material pairs.

Material Pair	Maximum Shear Stress (MPa)	Maximum Principal Stress (MPa)
(A) S355MC-Inconel^®^ 600	59.4 (magnetic layer, center)	137.3 (non-magnetic layer)
(B) S1100MC-Inconel^®^ 718	59.9 (non-magnetic layer)	116.5 (magnetic layer, center)
(C) S355MC-Inconel^®^ 718	37.3 (non-magnetic layer)	76.61 (magnetic layer, center)

## Data Availability

Data are contained within the article “Microstructure and mechanical properties of steel and Ni-based superalloy joints for rotors of high-speed electric motors.”.

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
