# Peer review of "Microstructure and Mechanical Properties of Steel and Ni-Based Superalloy Joints for Rotors of High-Speed Electric Motors"

_materials, 2022, doi:10.3390/ma15196906_

Round 1

Reviewer 1 Report

 Current paper studies var-ious dissimilar metallic joint, which tcan be used in high-speed ALA-SynRM rotors. Metallic joints of structural steels and Inconel® alloysare  fabricated by vacuum brazing and hot iso- 17 static pressing (HIP). Authors measure and analyze the shear strength, microstructure, hardness, and composition of the joints.  And the results are compared to FEA simulation. The paper is well-written, and can accepted in the journal.

Author Response

The authors thank the reviewer for the positive review and feedback.

Reviewer 2 Report

A very interesting article focused on the strength and structure of thee bimetallic join combinations. For the study, material pairs created with regard to the study of the possibilities of using high-strength steels for the design of high-speed rotors were used.

-The literature listed in the References is carefully cited in the article except of literature 33, which I did not find in the article (I apologize to the authors if I missed the citation)

- All abbreviations and symbols used are carefully explained in the article, which makes it easier for the reader to navigate the article

- Individual chapters are described clearly and comprehensively.

- The materials used and bonding processes are carefully and clearly described.

- The equipment and methods used are described clearly and comprehensibly. I recommend to mention the manufacturer Universal Testing Machine Matertest FMT-250.

- The results of the study are presented in figures/graphs and tables. The conclusions from the implemented experiments are discussed in detail.

 I recommend the work for publication.

Reviewer 3 Report

1. Abstract: chemical composition may be determined by EDS on SEM. You should indicate the detector used for chemical analysis.

2. Materials: The producers of used materials should be named (Company, Country).

3. Materials; First combination (i): S355MCD, not S355MC

4. Table 1: Please introduce CW021A in the text. Is it braze?

5. Diffusion bonding is usually done in vacuum. How can the HIP process lead to interfacial interaction in the presence of air? How does your process prevent oxidation at the interfaces at such high temperatures?

6. Line 193: “Shear stress can be evaluated by shaping”!?

7. Section 2.2:  Which software did you use for simulation? Ansys?

8. Table 5. Introduce “n” in the caption of the Table.

9. Figures 8-10: What is the mode of imaging? SE or BSE?

10. Line 366: How do you compare tensile and shear strengths?

11. Line 389: You are speaking about yield stress. The magnitude of strain determines the deformation value!

12. Lines 422-433: Indicate noticed phases in figures by arrows or…! You can say those are Ni-Al-rich phases. You can do some spot EDS to determine their molar composition.

13. In total: The element size for FEA is so coarse. If you use finer mesh, you will see higher stress values at the interfaces. You have not spoken about the fracture modes; were those interfacial or through the bulk? These kinds of studies also need fractography.

Round 2

Reviewer 3 Report

The corrections and responses to comments are almost convincing. My suggestion is to improve scientific English.

Comment 1: You can write :“The microstructure and chemical composition of the joints were studied by using optical microscopy, scanning electron microscopy (SEM) and energy dispersive spectroscopy (EDS) on SEM.”

Author Response

Author’s reply: The authors thank the reviewer for giving suggestion to improve scientific English.

Author's action: Text in lines 21-22 has been revised by the reviewer suggestions as following:

“The microstructure and chemical composition of the joints were studied by using optical microscopy, scanning electron microscopy (SEM) and energy dispersive spectroscopy (EDS) on SEM.